# Thermal light with enhanced oscillating bunching in field-tested photon-counting LIDAR

Theodor Staffas ✉, Jim Cleveborg , Jun Gao, Ali W. Elshaari & Val Zwiller

Traditional Light Detection and Ranging (LIDAR) systems rely on a modulated light source to encode information in the probe signal to measure distances. Alternatively, thermal LIDAR uses non-modulated thermal sources of bunched light. In this work, we demonstrate a bunched photon source at telecom wavelength, incorporated in a photon-counting LIDAR system, which determines distances by measuring the second-order correlation. The source is a sub-threshold laser spectrally filtered to extend coherence time, producing an oscillating $g^{(2)}$ curve with an enhanced bunching peak. Utilizing this oscillating bunching increases the signal-to-noise ratio and decreases the number of correlation events required to perform a measurement. This system achieves 2 ps resolution in both fiber-based and free-space measurements with up to 65 dB attenuation and ranges up to 87 km in optical fibers. Our approach demonstrates the possibility of increasing the bunching of classical light sources and improving their usability for LIDAR.

Since its conception in the 1960s[1], Light Detection and Ranging (LIDAR) has become a crucial tool for a diverse array of applications, ranging from consumer devices like smartphones and autonomous vehicles[2] to advanced research equipment in biomedical[3] and space exploration[4], not to mention in defense[5]. Traditional LIDAR systems rely on modulated light sources[6], which can be either intensity-modulated (i.e., pulsed lasers in direct detection systems)[7] or frequency-modulated (as in frequency-modulated continuous wave systems)[8,9]. This modulation results in a time structure in the reflected probe signal that enables a system to determine the time delay of the signal, and thereby the distance to an object[10]. An alternative approach that has gained more attention in recent years leverages non-modulated, time-correlated photons - commonly known as bunched light - to perform range measurements[11]. Using the statistical properties of photon arrival times, this method opens up new possibilities for LIDAR systems, enhancing both sensitivity and accuracy in a fundamentally different way. As this approach relies on the statistical properties of the source, measured by two independent detectors, it is inherently less sensitive to background noise mixing with the probe signal compared to traditional LIDAR systems[12]. Furthermore, since the probe signal

contains no inherent time-dependent signal trace that can be monitored for, it will effectively blend with natural thermal light from the surroundings, i.e., it is a new approach for stealth LIDAR.

The idea of leveraging photon bunching for sensing was first suggested and demonstrated by R. Hanbury Brown and R. Twiss in the 1950s when they pioneered a breakthrough technique to improve the angular resolution of astronomical measurements by observing the photon statistics of thermal light sources, such as stars[13]. At the heart of this method is the second-order correlation function, $g^{(2)}(\tau)$ defined by Eq. (1), which quantifies the likelihood of detecting two photons separated by a time delay, $\tau$. Where $I(t)$ is the intensity and $<>$ denotes the time average. This function peaks at zero time delay ($\tau = 0$) for certain sources (such as thermal sources), meaning photons are more likely to arrive in groups or 'bunches' at short intervals. This property, represented by a $g^{(2)}(\tau = 0) > 1$, reflects the chaotic nature of thermal light and is a distinctive feature of such sources[14]. Hanbury Brown and Twiss demonstrated this effect, now known as the HBT effect, in their famous measurement of the star Sirius's size using intensity interferometry[15]. By capitalizing on photon bunching, they showed that statistical correlations in light could

KTH Royal Institute of Technology, Department of Applied Physics, Albanova University Centre, Stockholm, Sweden. ✉e-mail: tstaffas@kth.se

reveal spatial information about distant objects.

$$g^{(2)}(\tau) = \frac{<I(t)I(t+\tau)>}{<I(t)>^2} \qquad (1)$$

More recent examples of bunched photons sensing applications include clock synchronization[16], ghost imaging, and range finding[11,17]. All these examples require a bright source of time-correlated photons. In this work, we use a subthreshold laser to generate bunched light in the near-infrared (1550 nm) and demonstrate how it can generate an oscillating bunching pattern, which increases the observed peak bunching and improves the signal-to-noise ratio, SNR, allowing for shorter integration times compared with using a conventional bunched source. The use of infrared photons is motivated by easier integration with telecommunication optical fibers and matching the atmospheric transmission window for free-space range measurements[18,19], taking advantage of lower atmospheric attenuation and lower solar noise[20]. We demonstrate the use of this source in a LIDAR system, performing measurements with a seconds integration time over 87 km distances in deployed optical fiber networks as well as in free space with 65 dB attenuation and 2 ps root-mean-square resolution.

There are two key parameters to consider for LIDAR measurements with a bunched light source: the level of bunching and the output power. The level of bunching, defined as the peak value of the second-order correlation, is directly related to the number of detected photon events required to distinguish the bunching peak from shot noise. Together, the bunching level and output power determine the integration time needed for accurate range measurements. In other words, two sources can be deemed equally effective for range-finding if the product of their bunching and output power is comparable.

Various techniques for generating bunched light have been explored: One approach employs spontaneous parametric down-conversion to create time-correlated photon pairs from a pump source, while another involves passing a continuous laser through a rotating ground glass plate to scramble the photon phase and produce pseudo-thermal light. Although these methods can achieve high bunching levels, their limited output power restricts practical application in LIDAR systems. In contrast, P.K. Tan et al.[21] demonstrated an elegantly simple yet effective design of bunched photons with significantly higher output power than any previously reported sources. This design, which we have adopted and built upon in our work, offers new potential for advancing LIDAR systems with enhanced efficiency and measurement range.

## Results

Our bunched photon source (illustrated in Fig. 1a), is based on a tuneable continuous wave (CW) external cavity laser diode (Toptica CTL 1550nm) operated just below the lasing threshold. In this operating mode, spontaneous emission dominates over stimulated emission, resulting in an output with a thermal nature. However, this emission is broadband – i.e., it will have a short coherence time– which negatively affects the second-order correlation. To increase the coherence time, the emitted light is coupled into a free-space filtering setup to reduce the emission spectral bandwidth. This filtering includes a 2 nm full-width at half-maximum (FWHM) bandpass filter (BPF) centered at 1549.5 nm (provided by Alluxa) and an etalon within a temperature control (TC) unit. The etalon is a 0.5 mm thick N-BK7 cavity (provided by Union Optic) with a refractive index of 1.5, Free Spectral Range (FSR) 197 GHz, and finesse 53, resulting in a 3.7 GHz FWHM transmission window. The CW laser cavity is tuned to 1549.5 nm to match as closely as possible the center of the BPF. The etalon has a temperature tuning coefficient of −1.4 MHz/K, making it resistant to minor lab temperature fluctuations; the TC component mitigates more significant, gradual drifts in the ambient temperature. Additionally, the transmission spectrum of the etalon can be fine-tuned by adjusting the incident angle, $\theta$, which shifts the center frequency of the etalon transmission peaks. This allows for precise alignment of an etalon transmission peak with the CW laser cavity. Figure 2a shows the measured unfiltered output of the thermal source as well as the transmission spectra of the etalon and BPF, and 2b display the filtered thermal output. The spectra shown in Fig. 2 were measured using a Yokogawa (AQ6370D) spectrum analyzer with 0.1 nm resolution.

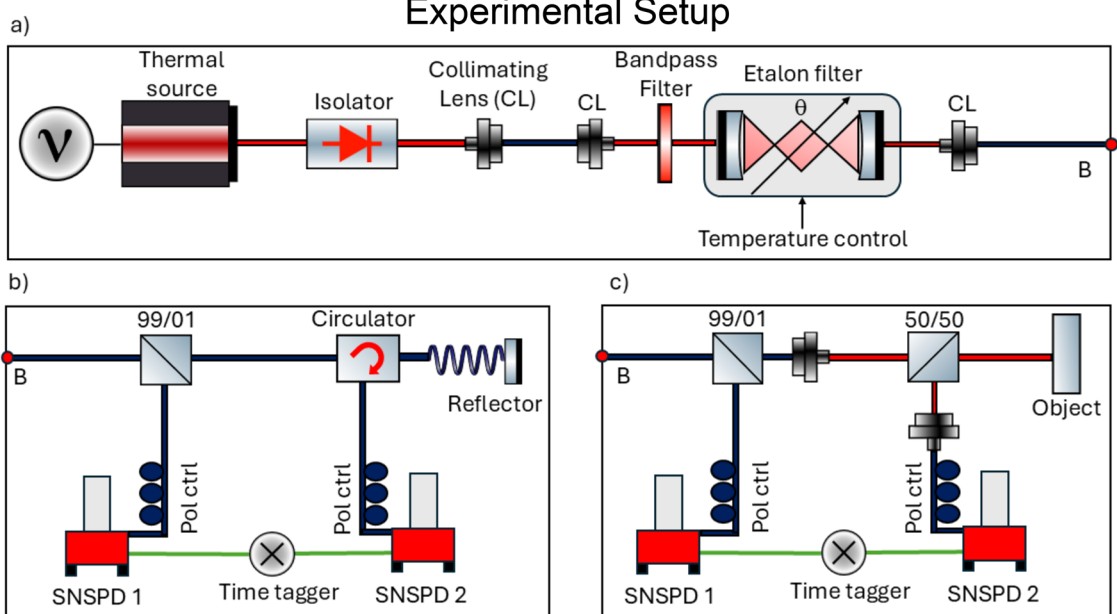

**Fig. 1 | Experimental setup. a** Source of bunched photons. The thermal source is an external cavity laser diode operated just below the lasing threshold. The output is filtered in a free-space setup by a bandpass filter and etalon combination. The etalon is placed in a temperature controller and $\theta$ is the incident angle of light through the etalon. **b** and **c** illustration of the fiber-based and free-space LIDAR setups used for measurements, respectively.

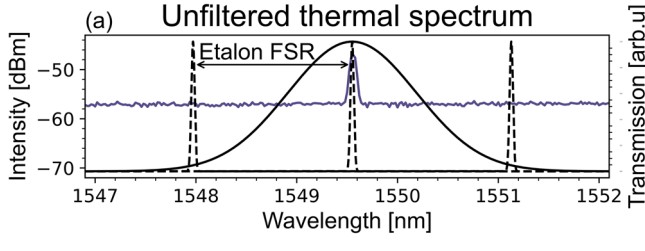

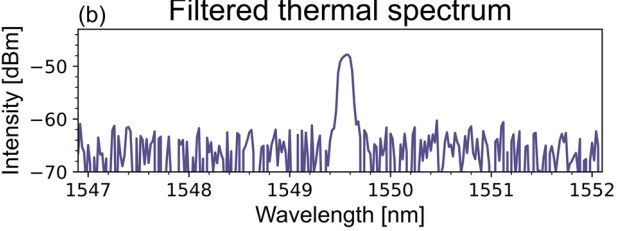

**Fig. 2 | Source output spectrum. a** Unfiltered output spectrum of the thermal light source (dark blue curve) with intensity displayed on the left y-axis. Transmission spectra of the bandpass filter (solid black line) and etalon (dashed black line) with transmission displayed on the right y-axis. **b** Fully filtered output spectrum of the thermal source, used for measuring.

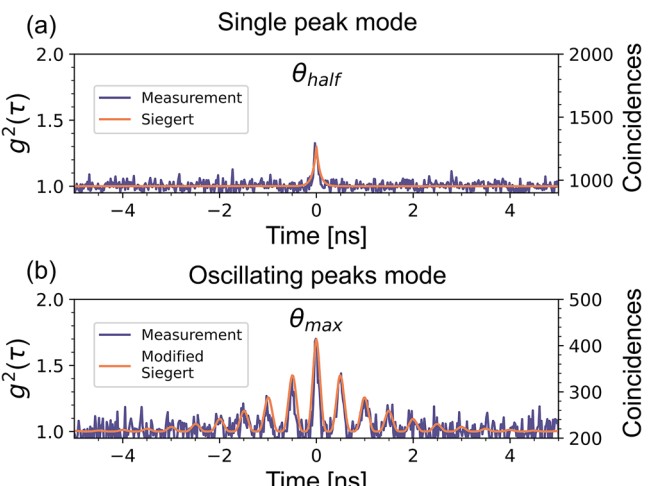

**Fig. 3 | Bunching measurements: Correlation curves measured using different etalon incident angles.** The left y-axis displays the second order-order correlation and the right axis displays the photon coincidences. **a** Measurement performed with the etalon at an arbitrary angle, $\theta_{half}$, producing a "classical" $g^{(2)}$ pattern displaying a single bunching peak and curve fit according to the Siegert relation in Eq. (2) with a coherence time of 0.1 ns. **b** Measurement performed at a very sensitive etalon transmission angle, $\theta_{max}$, producing an oscillating $g^{(2)}$ pattern displaying higher peak bunching, and curve fit according to the modified Siegert relation in Eq. (3) with a coherence time of 1.9 ns.

The source generates roughly $10^{10}$ photon events per second after filtering, and the output is coupled to the LIDAR system, either fiber-based as depicted in Fig. 1b) or free-space as depicted in Fig. 1c). In either case, the output is split into two paths. One path serves as a reference signal, directed to detector 1, while the second part is used as a probe signal, sent through a system where it may acquire a time delay before it is reflected back and coupled to detector 2. Both paths are routed through polarization controllers (Pol Ctrl) to ensure optimal polarization alignment before being coupled to two Superconducting Nanowire Single Photon Detectors (SNSPDs) (provided by Single Quantum). The use of SNSPDs in photon-counting LIDAR systems is advantageous due to their excellent performance at telecom wavelengths[22]. Our detectors have 80% detection efficiency at 1550 nm, with a timing jitter of ~20 ps, a maximum count rate of 10 million photon events per second, with no after pulsing, and approximately 100 dark counts per second. The detection events are recorded by a time-tagger (Swabian Timetagger X), enabling precise measurement of the second-order correlation function.

To demonstrate the bunching of our source, we performed correlation measurements with a zero-meter fiber delay between the two arms and different etalon transmission angles, $\theta$. The correlation events between SNSPD 1 and 2 are compiled in histograms with 20 ps binsize, chosen to match the timing jitter of the detectors. Figure 3 presents two correlation measurements for different values of $\theta$, with the second-order correlation displayed on the left y-axis and the number of correlation events displayed on the right y-axis. The angle $\theta_{max}$ represents the etalon incident angle at which the transmitted photon count rate is maximized; notably, this rate is twice as high as the rate for measurements performed at, $\theta_{half}$. However, adjusting the incident angle affects more than the detected photon countrate. At $\theta_{max}$ we observe a correlation curve with a higher $g^{(2)}$ value (1.75 compared with 1.25), a modulation in the coherence function (with 2 GHz frequency), and an extended decay time, indicating a longer coherence time—compared to the measurements taken at $\theta_{half}$.

Important to note is that there was not a single etalon transmission angle $\theta_{max}$ which produced the oscillating bunching pattern in Fig. 3b), but multiple angles. The oscillating bunching was observed whenever the etalon transmission angle caused the center of the etalon transmission window to perfectly align with the center of the laser cavity, at which point the measured count rate on both SNSPDs was doubled compared to the count rate at other transmission angles. However, this state was sensitive, and changes of $\pm 0.1^o$ caused the counter to half and the bunching pattern in Fig. 3a) returned.

This oscillating bunching behavior was not expected as the similar source used by P.K. Tan et al only measured bunching patterns similar to that in Fig. 3a)[21] (though they did not explore the correlation function as a function of etalon transmission angle). However, while unexpected, this oscillating second-order correlation is highly useful for LIDAR sensing as it enhances the observed bunching of the thermal source, therefore allowing measurements to be performed with fewer detected photons. This is demonstrated by the data in Table 1 where we present bunching measurements performed with different numbers of detected coincident photon pairs. Using both the single bunching peak and oscillating bunching mode, we attempt to measure the distance of a fixed fiber delay with decreasing average number of detected coincidences. For each setting, we perform 50 measurements per coincidence rate and calculate the spread in the measured time delay, where $q_{75} - q_{25}$ is the difference between the 25th and 75th percentiles of the measurements.

These measurements demonstrate that the enhanced bunching pattern in Fig. 3b) enables measurements with fewer detected

**Table 1 | Measurement sensitivity: Statistics of 50 LIDAR measurements using the single bunching peak mode and the oscillating peak mode operation of the thermal source as in Fig. 3**

| Coin. single peak | 430 | 220 | 150 | 80 |
|---|---|---|---|---|
| $q_{75}-q_{25}$ [ps] | 15 | 40 | 55 | 12 340 |
| Coin. oscillating peaks | 480 | 180 | 90 | 60 |
| $q_{75}-q_{25}$ [ps] | 0 | 20 | 20 | 55 |

The value of $q_{75} - q_{25}$ is the difference between the 25th and 75th percentile which we define as the accuracy of the measurements.

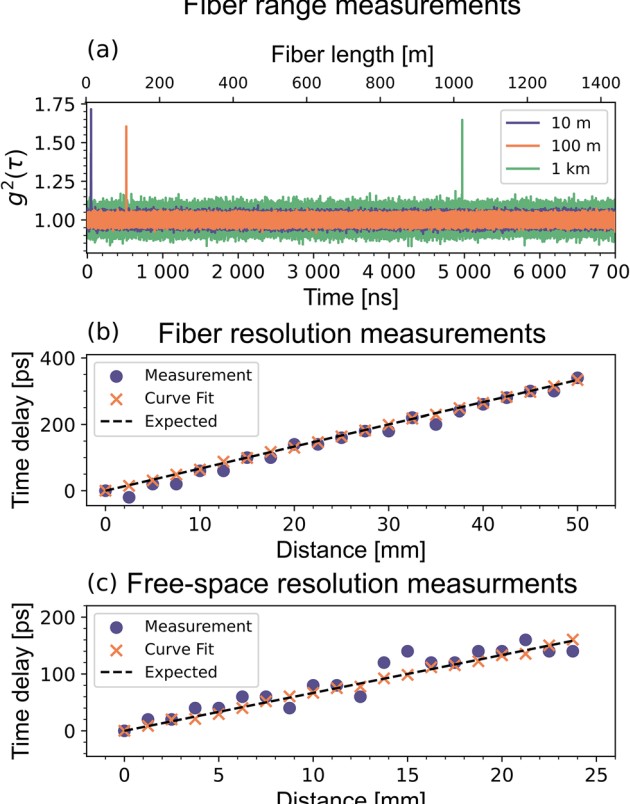

**Fig. 4 | Range and resolution measurements. a** Range measurements using fiber spools as delay lines. **b** Range measurements with variable fiber delay. **c** Range measurements in free space using a translation stage as a variable delay. Circles correspond to peak values from the histograms and crosses correspond to curve fitting using Eq. (3). These can be compared to the expected theoretical values given by the dashed line.

correlation events compared with measurements using the non-oscillating bunching.

The underlying cause of this oscillating bunching behavior and the difference in coherence time of the two bunching curves displayed in Fig. 3 remains an open question at this point and trivial explanations such as internal reflections in the experimental setup corresponding to the oscillation period can easily be ruled out. The behavior is highly interesting and warrants further research in its own right, and will be addressed in future work, with a detailed theoretical model under development. For now, we can conclude that the results shown in Table 1 clearly demonstrate its advantage in LIDAR applications. Since it enables measurements with shorter integration times through systems with higher signal losses. Therefore, all other measurements presented in this paper are performed using the source in this oscillating bunching mode.

However, there is still one clear limitation using the source in the oscillating bunched mode vs the single peak mode: the lack of a mathematical model of the $g^{(2)}$ curve. The "classical" bunching curve demonstrated in Fig. 3a) follows the well-known Siegert relation, described by equation (2)[23].

$$g^{(2)}(\tau) = 1 + \beta|g^{(1)}(\tau)|^2 = 1 + \beta(e^{-\frac{|\tau|}{\tau_c}})^2 \qquad (2)$$

Where $\beta$ is a parameter defining the level of bunching, $\tau$ is the time delay between two photons, and $\tau_c$ is the coherence time of the source, which is inversely proportional to the spectral width[23]. As demonstrated by the orange curve fit, the correlation curve measured in

Fig. 3a) aligns well with this model, with a measured coherence time of roughly 0.1 ns.

However, the measured bunching in Fig. 3b) does not align with the Siegert relation, which presents a challenge. A mathematical model of the expected $g^{(2)}(\tau)$ is highly useful for curve fitting of the measured data in order to improve the accuracy of LIDAR measurements. We therefore propose a "modified" Siegert relation, equation (3), to describe the oscillating bunching behavior seen in figure 3b).

$$g^{(2)}(\tau) = 1 + \beta(e^{-\frac{|\tau|}{\tau_c}})^2 (1 + \cos(2\pi\Delta\nu\tau)) \qquad (3)$$

The main difference between the two relations is the factor $(1 + \cos(2\pi\Delta\nu\tau))$, which models the oscillations in the $g^{(2)}$ curve, where $\Delta\nu$ is a parameter describing the oscillation frequency, and the orange curve fit in figure 3b) shows that this modified Siegert relation accurately aligns with the experimentally observed oscillatory bunching behavior. Using Eq. (3), we extract the measured coherence time of the of bunching pattern in figure 3b) as roughly 1.9 ns and the oscillating frequency $\Delta\nu \approx 2$ GHz. Using this model to curve fit the measured data, we can improve the accuracy and stability of our LIDAR measurements.

To further evaluate the usability of this source and oscillating bunching for LIDAR measurements, we investigate the maximum range and the resolution achievable. Unlike more traditional LIDAR systems that utilize pulsed lasers, which notoriously suffer from problems with range ambiguity stemming from the deterministic emission of the probe signal, our approach—based on measuring the second-order correlation function—does not have a theoretical range limit. This advantage arises because the emission from the thermal source used in this setup is inherently random, thereby eliminating range ambiguity. The maximum range will therefore only depend on the experimental conditions, such as source brightness and level of bunching, as well as the losses of the probe signal. To investigate the maximum range of the system, we performed correlation measurements using 3 different fiber delays, 10 m, 100 m, and 1 km. The resulting $g^{(2)}$ curves, displayed in Fig. 4a), were obtained under identical integration times and comparable photon count rates. Notably, the bunching signal has no significant decay across these delays.

Furthermore we wish to investigate the resolution of the system, which is constrained by the total timing jitter, $\tau_{j_{tot}}$, which in turn is dependent on the timing jitter of individual components, $\tau_{j_{component}}$, as defined in Eq. (4). The relevant components include the two photon detectors and the two channels of the timetagger, with timing jitters of 20 ps and 2 ps, respectively. It's also possible to include the effects of dispersion in this expression. However, since this can be compensated for as demonstrated in[24], we can safely ignore these effects in these measurements. This results in a total timing jitter of approximately $\tau_{j_{tot}} = 28$ ps, corresponding to a roundtrip distance resolution of roughly 5 mm in free space. This limits the resolution achievable by using the peak of the measured $g^{(2)}$ curve.

$$\tau_{j_{tot}} = \sqrt{\sum \tau_{j_{component}}^2} = 28\,\text{ps} \qquad (4)$$

To surpass this resolution limit, we apply curve fitting of equation (3) to the measured $g^{(2)}$, allowing for a more precise determination of the time delay. We investigate the resolution achievable both in fiber-based and free-space measurements using a variable fiber delay and a translation stage, respectively. The results are shown in Fig. 4b and 4c, where the time delays determined using the peak $g^{(2)}$ value are displayed as dots while the time delays determined using curve fitting are displayed as crosses and can be compared to the expected value given by the dashed line. Comparing the difference between the values obtained by curve fitting with the expected values we achieve a root-mean-square error of 2 ps for both free-space and fiber-based measurements, corresponding to 0.3 mm roundtrip distance in free space.

### (a)

Fiber network map

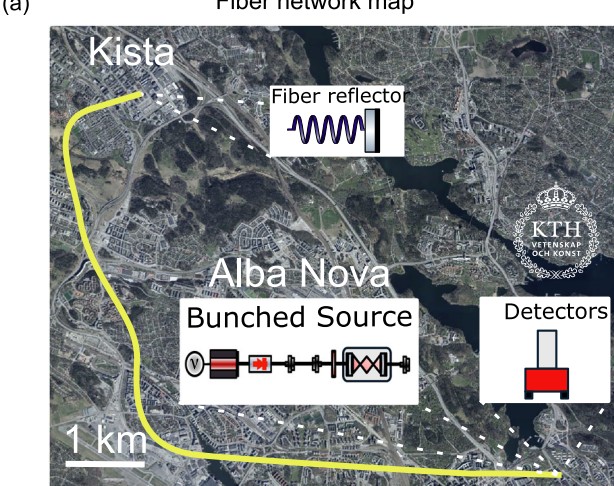

Fiber network measurement

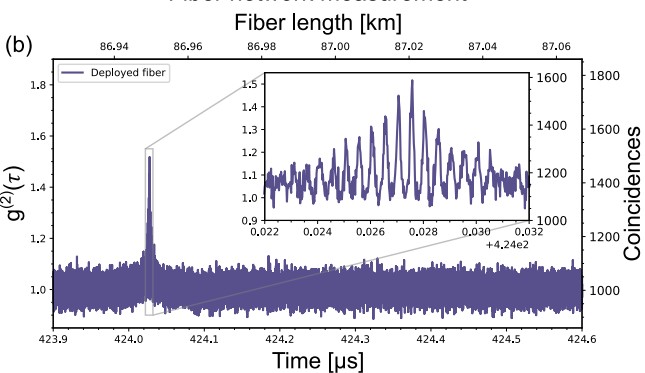

**Fig. 5 | Measurements in deployed optical fibers. a** Map of the deployed fiber network between KTH at campus Alba Nova and Ericsson Research lab at Kista, this map is only an illustration of the fiber network; the exact route is classified. **b** Range measurement in an 87 km fiber network (40 km deployed optical fiber and 47 km additional fiber spools) with 40 dB losses and 30,000 counts/second background noise from crosstalk.

Lastly, we use our system to measure the fiber delay of an optical fiber network deployed across the Stockholm metropolitan area, see Fig. 5a, totaling 87 km in length with roughly 40 dB attenuation. Using a deployed optical fiber also introduced a background level from crosstalk significantly higher than for measurements using only fiber spools in our lab, rising from approximately 100 counts/second to 30,000 counts/second. Despite this, as illustrated in Fig. 5b, the bunching signal exhibited no significant decay, even under these detrimental conditions.

## Discussion

In this work, we have demonstrated a high-brightness source of bunched photons (~$10^{10}$ photons per second) using a sub-threshold laser as a thermal source and spectrally filtering the output to increase the coherence time. This source, operating within the telecom C-band, is ideally suited for integration into optical fiber networks and free-space measurements, benefiting from alignment with the atmospheric transmission window. Moreover, we have shown the possibility of this source to produce an oscillating second-order correlation which amplifies the observed peak bunching, increasing the peak $g^{(2)}$ from 1.25 to 1.75, and doubling the source brightness. This increased peak bunching significantly reduces the number of photon correlation events required for LIDAR measurements, as demonstrated by the data in Table 1. Using our setup, we achieve 2 ps (i.e., mm-level) range

resolution in both fiber-based and free-space measurements and demonstrate bunching measurements across a total distance of 87 km in fibers, as demonstrated by the data in Fig. 5, underscoring the potential for scalable, high-precision distance measurements in both terrestrial and fiber-optic environments.

## Methods
### Oscillating bunching measurements
To obtain the measurements presented in Fig. 3a the thermal source was coupled to a spectral filtering setup, and the transmission angle of the etalon was tuned whilst observing the countrate. The count rate remained largely unchanged except for specific angles when the transmission peak precisely overlapped with the center of the laser cavity, at which point the count rate sharply doubled. This state was sensitive to changes on the scale of ±0. 1° ° and if moved beyond this value, the count rate on both detectors dropped by half again. The second-order correlation was then measured for both this specific transmission $\theta_{max}$ where the countrate was doubled, and for the more arbitrary angle $\theta_{half}$, which was any other transmission angle.

### Statistical measurement
Using a fixed fiber delay between the two arms, we measure the second-order correlation using both the conventional and the oscillating $g^{(2)}$ curves and extract the coincidence counts and time delay at the peak correlation. By adjusting the integration time and attenuation of the thermal source we control the number of recorded coincidences in each measurement. For each setting tested, we perform 50 measurements, extracting the mean measured time delay corresponding to a specific coincidence count and calculating the difference between the 75th and 25th percentile ($q_{75}$ and $q_{25}$), using these metrics reduces the impact of outliers in the data compared to using the average and standard deviation.

### Resolution measurements
For the fiber-based resolution measurements, a variable fiber delay (Newport VariDelay$^{TM}$) was used to control the time delay between the two arms of the setup, and the fiber delay was swept from 0 to 50 mm using 2.5 mm steps, totaling 20 separate measurements. For the free-space measurements, a mechanical translation stage was used to control the time delay and the stage was swept from 0 to 25 mm in 1.25 mm steps, which equates to a change in round-trip distance of 2.5 mm.

The integration time of these measurements was 20 seconds for the fiber-based measurement and 300 seconds for the free-space measurement. The longer integration time for the free-space measurements was due to the high attenuation of the probe signal, 65dB. To compensate for the weak detected probe signal, the reference detector was operated near saturation levels to maximize the measured coincidence rate, thereby reducing the integration time needed.

### Fiber network measurements
The deployed optical fiber link used extended between our lab at KTH campus Alba Nova to a lab at Ericsson Research Lab in Kista. The fiber network is roughly 20 km in distance, and a partial fiber reflector at the second node reflects part of the signal back through the network, doubling the fiber delay. We also added 47 km of fiber spools in our lab, resulting in an 87 km fiber delay with roughly 40 dB attenuation (accounting for the partial reflector), which is much higher compared to the expected 0.2 dB/km loss in an ideal optical fiber. The integration time of the measurement was 30 seconds, with reference signal powers comparable to those used in the fiber-based resolution measurements.

## Data availability
The data generated in this study is available at the following repository: https://github.com/tstaffas/Thermal-LIDAR.git[25].

## Code availability

All code to generate the figures presented in this manuscript is available at the following repository: https://github.com/tstaffas/Thermal-LIDAR.git,[25].

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

## Acknowledgements

We wish to thank Gemma Vall-Llosera at Ericsson AB for allowing us to access the deployed fiber network between our lab at KTH and their facility in Kista. We also wish to thank P.K. Tan and K. Mølmer for their helpful discussions about the project. Lastly, we acknowledge support from the European Union's Horizon 2020 Research and Innovation Action under grant agreement no. 899824 (FET-OPEN, SURQUID) (V.Z.).

## Author contributions

The initial idea for the project was conceived by V.Z. and T.S. The experimental setup was constructed by T.S. and J.G. Measurements, data analysis, and modeling were performed by T.S. and J.C. with supervision by V.Z. and A.E. The manuscript was written by T.S. with input from all authors.

## Funding

## Competing interests

The author Val Zwiller is a co-founder of the company Single Quantum, which provided the detection system used in this project. Others declare no conflicts of interest.
