## [Transparent Peer Review file · Nature Communications]

Thermal light with enhanced oscillating bunching in field-tested photon counting LIDAR

Corresponding Author: Dr Theodor Staffas

Version 0:

Reviewer comments:

Reviewer #1

(Remarks to the Author)

This manuscript presents a LIDAR system using bunched photons at telecom wavelengths, achieved by leveraging interference between two longitudinal modes of a sub-threshold laser. The approach enhances signal quality and allows for high-resolution, long-range distance measurements with photon-counting LIDAR technology.

I believe the main issue with this manuscript is that the proposed LIDAR system is demonstrated in optical fibers, rather than in free space. Free space would introduce significantly more complexity, such as higher attenuation coefficients. Additionally, factors like atmospheric aerosols, wind speed, and turbulence would have a more pronounced effect, which could impact the performance of the LIDAR system. Therefore, if the authors can provide a demonstration in free space, this manuscript would be more suitable for publication.

The manuscript uses two FPIs to filter the two longitudinal modes of the sub-threshold laser. Would filtering more longitudinal modes further improve the detection range?

Figure 5 shows the two longitudinal modes. How can the authors confirm whether additional longitudinal modes are involved?

The impact of fiber dispersion should be analyzed. Specifically, it is important to determine whether the dispersion effects exceed the 2 ps temporal resolution mentioned in the manuscript.

The maximum counting rate of the SNSPDs is 10 M. Does this specific value impact the system's performance? Further analysis is required.

The manuscript demonstrates a detection range of up to 87 km in optical fibers. What is the maximum achievable range for this system? Can the authors provide a derivation or further justification?

Reviewer #2

(Remarks to the Author)

This manuscript utilizes interference of two non-coherent longitudinal modes to enhance bunching and output power for the photon-counting LIDAR. The results demonstrate a sub-mm-level accuracy and 87-km with 40-dB losses distance measurement, which are outstanding. However, the innovation is insufficient which the basic idea and setup for this research was developed from the Ref. 20. Therefore, the current manuscript does not fulfill the criteria for Nature Communications. I cannot recommend this manuscript in the current format mainly according to the following comments.

- 1 What main different between this manuscript and Ref. 20? Why do similar setup result in such significant differences in results. Can the existing results be achieved using the method in the Ref. 20? If not, what is the reason.
- 2 What advantages of this technique in the distance ranging? The main advantage of photon-counting ranging is the high sensitivity, but the commercial OTDR or other ranging techniques also can achieve 40-dB dynamic range or more.
- 3 The two longitudinal modes output spectrum is more meaningful to demonstrate than the filtered one in Fig. 2b). Why is

there no figure in the manuscript.

4 Please explain more about the reason of the interference effect of two non-coherent longitudinal modes. How to maintain interference characteristics in long-distance transmission

5 The conditions for spatial transmission are significantly different from those in optical fibers. There are only the results of test in fiber, and it is inappropriate to mention free-space measurement.

Reviewer #3

(Remarks to the Author)
See the attached pdf.

Version 1:

Reviewer comments:

Reviewer #2

(Remarks to the Author)
All responses explained my questions. The changes and supplementary material completed the content of the manuscript. Based on the comments of other reviewers and the corresponding responses, this manuscript can be accepted.

Reviewer #3

(Remarks to the Author)
See attached pdf. Please respond to the two main questions raised.

Version 2:

Reviewer comments:

Reviewer #3

(Remarks to the Author)
I appreciate the new supplemental material provided by the authors, which corrects inconsistencies in the previous version.

However, I still have qualms about the response to my second question in the previous review (“I miss a physical explanation of the reason for the extended decay time...”). Letting aside that I don’t understand the explanation given in the rebuttal (which could be my fault), I think that equation (3) using the same symbol τ_c for the coherence time as equation (2), without more explanation, is very misleading. Obviously, the width of the very narrow correlation peak shown in fig. 3 (a) must be dominated by τ_c , which must correspond to a very short time. But if that same τ_c is used in Eq. (3), it won’t yield the second order correlation shown in fig. 3(b) (the correlation peaks for would be swamped in the noise). Therefore, whatever the reason, τ_c must have increased (a lot) between the value used to obtain the curve fit of fig. 3(a) and that used for the curve fit of fig. 3(b).

I think that everything would be easier to understand if the parameter values of the fitting model (among which τ_c) were given in the text. I agree with the authors that “the main point of this work is not to be a theoretical derivation of the bunching behaviour [...]. Our goal with the model was to be able to improve the accuracy of the measurements [...]”; nevertheless, they do provide a model (Eqs. (2) and (3)), and this should at least be consistent. If no satisfactory explanation can be given for the increase of the coherence time, it would be better declaring right away that the “theoretical derivation of the bunching behaviour” is not yet fully explained and that it “warrants more research in its own right”.

I’m sorry, but I cannot recommend the paper publication until the parameters of the model leading to the fittings shown in fig. 3 are given.

(Remarks on code availability)

Version 3:

Reviewer comments:

Reviewer #3

(Remarks to the Author)
See pdf.

(Remarks on code availability)

The code seems to be intended to generate the manuscript figures. It fails for some figures because a Python module is missing. No fundamental code seems to be available, nor is needed within the scope of the paper.

Changes to the manuscript:

- (1) The LIDAR acronym has been updated to the correct meaning “Detection” as opposed to “Distance”.
- (2) The use case of LIDAR for atmospheric sensing has been removed as it is not clear the proposed LIDAR system would be useful for such measurements. Line 5->7.
- (3) The text in lines 22 -> 30 has been updated to clarify the advantage of stealth and tolerance from background noise.
- (4) Figure 1 has been updated for readability and subfigures 1b) and 1c) have been added to clarify the two use cases of fiber vs free-space measurements. The figure text has also been updated correspondingly.
- (5) The text in lines 139-147 has been updated according to the changes in figure 1.
- (6) The text in lines 217->222 has been updated to clarify that the absence of a direct measurement of the multiple laser modes in the probe signal is due to the limited resolution of our spectrum analyzer.
- (7) The text in lines 256 -> 275 has been updated to discuss the theoretical infinite range of a bunched LIDAR system.
- (8) Figure 4c) was added with free-space resolution measurements performed with -65 dB attenuation of the probe signal, demonstrating the same resolution as achieved in the fiber-based measurements.
- (9) The text in lines 293->314 was updated to describe the additional results presented in figure 4c).
- (10) The text in lines 286-288 has been updated to mention that the dispersion effect can be included in the RMS value of the total timing jitter, but that this can also be compensated for.
- (11) The text in lines 315 -> 324 has been updated to include a brief discussion of the benefit of using a detector with a high maximum countrate.
- (12) Figure 5a) was updated to reflect the changes in figure 1, and figure 5b) was updated with a right y-axis similar to those displayed in figure 3. The figure text has also been updated correspondingly.
- (13) Addition of a supplementary material containing the derivation of the modified Siegert relation presented in equation 3.
- (14) A grammatical check of the manuscript was performed, and minor changes were implemented.
- (15) The two references cited in this response has been added to the manuscript as well.
- (16) All data presented in the figures, as well as the code to reproduce the figures, have been uploaded to an online repository with open access. The link to the repository has been added to the manuscript

Reviewer #1 (Remarks to the Author):

This manuscript presents a LIDAR system using bunched photons at telecom wavelengths, achieved by leveraging interference between two longitudinal modes of a sub-threshold laser. The approach enhances signal quality and allows for high-resolution, long-range distance measurements with photon-counting LIDAR technology.

- I believe the main issue with this manuscript is that the proposed LIDAR system is demonstrated in optical fibers, rather than in free space. Free space would introduce significantly more complexity,

such as higher attenuation coefficients. Additionally, factors like atmospheric aerosols, wind speed, and turbulence would have a more pronounced effect, which could impact the performance of the LIDAR system. Therefore, if the authors can provide a demonstration in free space, this manuscript would be more suitable for publication.

Response: We thank the reviewer for this comment. Following the reviewer's suggestion, we have conducted new resolution measurements using a free-space setup and present the new results in figure 4c). These measurements were more challenging due to the higher attenuation of the probe signal, -65dB, but the resulting resolution was the same as for the fiber-based measurements.

Unfortunately, we do not have access to a free-space LIDAR system outside our laboratory and therefore we were unable to address the question regarding wind and turbulence. In general this will introduce additional losses of the probe signal and will present a challenge to the emission and collection optics used, rather than the technique it-self. The main challenge of a LIDAR technique is its ability to operate under photon-starved conditions and this we demonstrate in the measurements in figure 4c) as well as the measurements in table 1.

Changes to manuscript: Figure 4c) was added with free-space resolution measurements performed with -65 dB attenuation of the probe signal, demonstrating the same resolution as achieved in the fiber-based measurements. The text in lines 285->311 was updated to describe the additional results presented in figure 4c). Figure 1 has also been updated and in particular figure 1c) present the experimental setup of the free-space measurements.

- The manuscript uses two FPIs to filter the two longitudinal modes of the sub-threshold laser. Would filtering more longitudinal modes further improve the detection range?

Response: This is an interesting question. First, a clarification: in the manuscript we use a single Fabry-Perot Interferometer in combination with a Bandpass Filter to filter out two longitudinal laser modes.

Now for the question: "*Would filtering more longitudinal modes further improve the detection range?*". Using the current setup we are not able to filter an arbitrary amount of modes from the laser. The limitation is set by the FWHM of the etalon transmission peak. We can state that there must be a limit where a large number of modes simply start behaving as a single mode with a broader spectrum and shorter coherence time, which is detrimental to LIDAR measurements. This evident by the fact that measurements using only the bandpass filter, which transmits a great number of modes of the laser cavity, does not exhibit any bunching. This is demonstrated in figure 1 below.

What the optimal number of modes would be warrants detailed theoretical and experimental analysis, but is outside the scope of this project.

Changes to the manuscript: We have included our derivation of the modified Siegert relation in a supplementary material which can serve as the basis for a numerical simulation.

Figure 1: Two LIDAR measurements performed using a 5 m fiber delay. a) Filtering using only a BPF. b) Filtering using both BPF and etalon. No bunching is observed in the unfiltered measurement despite the higher number of recorded coincidences.

- Figure 5 shows the two longitudinal modes. How can the authors confirm whether additional longitudinal modes are involved?

Response: This is an excellent point raised. Due to the limited resolution of our spectrum analyzer, we are not able to directly measure the two longitudinal modes using our spectrum analyser as the mode spacing is smaller than the resolution. From curve fitting equation 3 to the measurement displayed in figure 3b) in the manuscript, the mode spacing is determined to be approximately 2 GHz (~16pm) and the resolution of our spectrometer is 1 nm.

To address the reviewer's question we contacted the manufacturer of the laser used (Toptica) and asked them about the mode spacing of the laser. Their response (slightly lacking in detail) was: "*The free spectral range of the [laser model] is determined by the resonator with a length of centimeters. But we cannot share more details than this*". They are not willing to share exact information of the mode spacing but they confirmed that the resonator size is cm-scale which would give free spectral range of GHz-range, and since the FWHM of our etalon is only 3.7 GHz, it would be unlikely that we can have multiple modes from the laser in our measurements.

The main reason we are confident in stating that there are only two modes involved is that the measurement results are well described by the modified Siegert relation, which we have derived under the assumption of only two mode interference.

Changes to manuscript: The text in lines 214->219 has been updated to clarify that the absence of a direct measurement of the multiple laser modes in the probe signal is due to the limited resolution of our spectrum analyzer.

- The impact of fiber dispersion should be analyzed. Specifically, it is important to determine whether the dispersion effects exceed the 2 ps temporal resolution mentioned in the manuscript.

Response: We appreciate this insightful comment. The dispersion coefficient of an SMF-28 optical fiber is approximately 18 ps/(km·nm). Given a mode spacing of 2 GHz (~16 pm), the dispersion-induced broadening can be estimated as ~0.29 ps/km.

This broadening can be incorporated as a term into the system's total timing jitter, as described by equation 4. However, since this is an RMS value, it will be dominated by the largest contributing term—the timing jitter of the detectors. In our system, the detector jitter remains the primary limiting factor, and only at significantly longer distances would dispersion effects become dominant.

For reference, at 87 km, dispersion would contribute ~25 ps of broadening, which is on the same order as the timing jitter of the detectors. However, it is important to note that our source is a continuous-wave source, meaning that chromatic dispersion is unlikely to significantly affect the temporal resolution of the measurement. Given the maximum fiber delay available in our setup, no discernible impact due to dispersion was observed in our experimental results. Figure 4 further supports this, demonstrating that the system's temporal resolution is not fundamentally limited by dispersion effects.

Thus, while fiber dispersion contributes to the overall timing uncertainty, its impact in our specific setup remains subdominant compared to detector jitter. For longer measurements it may prove necessary to include dispersion compensating elements such as those demonstrated in [1].

Changes to manuscript: The text in lines 286-288 has been updated to mention that the dispersion effect can be included in the RMS value of the total timing jitter, but that this can also be compensated for.

- The maximum counting rate of the SNSPDs is 10 M. Does this specific value impact the system's performance? Further analysis is required.

Response: We thank the reviewer for this comment. In short, the number of coincidences per unit time is dependent on the product of the countrate (ctr) on the two detectors, i.e. coincidences $\propto \text{ctr}_1 * \text{ctr}_2$. Therefore, a higher maximum countrate will decrease the required integration time of a measurement.

Of course, in a LIDAR measurement the probe signal is often heavily attenuated by several factors such as scattering and absorption in the optical path and therefore the probe signal will not be close to the maximum countrate of the detector. However, the reference signal is not attenuated and may be powerful enough to saturate the reference detector. Therefore, it is advantageous to have a high maximum countrate of the reference detector to allow for a stronger reference signal.

Changes to manuscript: The text in lines 314 -> 322 has been updated to include this information.

- The manuscript demonstrates a detection range of up to 87 km in optical fibers. What is the maximum achievable range for this system? Can the authors provide a derivation or further justification?

Response: This is an excellent question: What is the theoretical maximum range of this system? Firstly, since the system measures the second-order correlation of a thermal source, whose emission is

random by nature, there is no problem of maximum unambiguous range. Therefore, one answer to this question is: there is no maximum distance to a thermal LIDAR system given enough integration time.

This is in contrast with more traditional LIDAR systems that use pulsed light as a probe signal, that has a maximum range determined by the repetition rate of the laser pulses. It is of course possible to extend this maximum range by for example using pseudo-randomly spaced laser pulses but this introduces more complexity in the system.

However, our system will have a “practical” maximal range beyond which the probe signal is too weak to efficiently determine the distance. This is dependent on parameters such as source brightness and level of bunching, as well as collection efficiency of the LIDAR system. Table 1 is an attempt to quantify this as we show that using our setup, we only require 180 coincidences to reliably measure a bunching peak and thereby determine the time-delay. This number should be independent of the distance.

Changes to manuscript: The text in lines 253 -> 273 has been updated to discuss the theoretical infinite range of a bunched LIDAR system.

References:

[1] Liu, Shujun, et al. "Ultra-compact thin-film-lithium-niobate photonic chip for dispersion compensation." *Nanophotonics* 13.26 (2024): 4723-4731.

Reviewer #2 (Remarks to the Author):

This manuscript utilizes interference of two non-coherent longitudinal modes to enhance bunching and output power for the photon-counting LIDAR. The results demonstrate a sub-mm-level accuracy and 87-km with 40-dB losses distance measurement, which are outstanding. However, the innovation is insufficient which the basic idea and setup for this research was developed from the Ref. 20. Therefore, the current manuscript does not fulfill the criteria for Nature Communications. I cannot recommend this manuscript in the current format mainly according to the following comments.

I What main different between this manuscript and Ref. 20? Why do similar setup result in such significant differences in results. Can the existing results be achieved using the method in the Ref. 20? If not, what is the reason.

Response: This is an interesting question. We would highlight that the main differences between the work in this manuscript and the work by Tan et al in Ref. 20 is:

- We show that we can use two modes to improve the bunching compared to using a single mode. This is demonstrated in table 1 where we show that we require fewer coincidences to determine the bunching peak when measuring two modes compared with measuring a single mode.

Our ability to use multiple modes is dependent on the mode spacing of the external cavity laser diode we used, as well as the specifics of the etalon and bandpass filter. We cannot determine if this could be performed with the diode used in Ref. 20.

- We shifted the operating wavelength from the visible spectrum to the near infra-red to take advantage of the atmospheric transmission window and lower solar noise, see figure 1 in [2], to push the ability to use this LIDAR method in free-space measurements. Furthermore it facilitates easier integration in optical fiber networks that predominantly rely on telecommunication wavelengths in the near infra-red.
- We demonstrate a resolution 4 times better than that demonstrated in ref 20 (2 ps vs 8 ps).
- We demonstrate the use of SNSPD detectors with higher detection efficiency and lower timing jitter than those used by Tan et al.

2 What advantages of this technique in the distance ranging? The main advantage of photon-counting ranging is the high sensitivity, but the commercial OTDR or other ranging techniques also can achieve 40-dB dynamic range or more.

Response: We believe there are three distinct advantages to LIDAR using thermal sources that we try to highlight in the manuscript.

- 1: The source required for this LIDAR technique is much “simpler” than the modulated laser sources used in more traditional LIDAR systems. i.e. the source requires less advanced components which can reduce the cost of the system.
- 2: As discussed in a previous answer to reviewer 1 this type of LIDAR system has theoretically infinite unambiguous range due to the random emission of the thermal source. This contrasts with modulated LIDAR systems that have problems of ambiguous distances for pulsed LIDAR systems or limitations on detector bandwidth for coherent LIDAR systems.
- 3: As briefly mentioned in lines 26-28 of the manuscript, this LIDAR method imparts no time-dependent trace in the probe signal that is discernible from background signals, making it an interesting approach for “stealth” LIDAR. In other words, because the probe signal is simply thermal light it will blend in with other thermal light from the environment and therefore it is harder to monitor for surveillance.

Changes to the manuscript:

The text in lines 27 -> 30 has been updated to clarify the advantage of stealth and tolerance from background noise.

The text in lines 253 -> 273 has been updated to discuss the theoretical infinite range of a bunched LIDAR system.

3 The two longitudinal modes output spectrum is more meaningful to demonstrate than the filtered one in Fig. 2b). Why is there no figure in the manuscript.

Response: We thank the reviewer for this comment, this is similar to a previous comment by reviewer 1. We agree that it would be very meaningful to directly measure the two modes but we are limited by the resolution of the spectrum analyser, 1nm, and the mode spacing of approximately 2GHz ~ 16 pm.

Changes to the manuscript: The text in lines 214->219 has been updated to clarify that the absence of a direct measurement of the multiple laser modes in the probe signal is due to the limited resolution of our spectrum analyzer.

4 Please explain more about the reason of the interference effect of two non-coherent longitudinal modes. How to maintain interference characteristics in long-distance transmission

Response: We thank the reviewer for this comment. In order to provide a more thorough explanation of the origin of the interference of the two longitudinal modes we provide our derivation of equation 3 in the supplementary material.

Here we use an assumption of two Lorentzian shaped modes and use the Wiener-Khinchin theorem to relate the spectrum to the first-order correlation, $g^{(1)}$. We then use the original Siegert relation to derive the expression of the second-order correlation expressed in equation 3 of the manuscript.

This derivation relies solely on the spectrum of the bunched light and as long as this spectrum is maintained over the long-distance transmission the interference characteristic should remain.

5 The conditions for spatial transmission are significantly different from those in optical fibers. There are only the results of test in fiber, and it is inappropriate to mention free-space measurement.

Response: This is an excellent point raised. Following the reviewer's suggestion, we have conducted new resolution measurements using a free-space setup and present the new results in figure 4c).

From these measurements we conclude that the 2 ps resolution is the same for free-space measurements as well, despite the 65 dB of attenuation of the probe signal compared to the fiber-based measurements.

Changes to manuscript: Figure 4c) was added with free-space resolution measurements performed with -65 dB attenuation of the probe signal, demonstrating the same resolution as achieved in the fiber-based measurements. The text in lines 285->311 was updated to describe the additional results presented in figure 4c).

References:

[2] Liao, Sheng-Kai, et al. "Long-distance free-space quantum key distribution in daylight towards inter-satellite communication." *Nature Photonics* 11.8 (2017): 509-513.

Reviewer #3 (Remarks to the Author):

The manuscript reports a system for ranging with very high precision using the correlation function of intensity fluctuations (second-order correlation) of pseudothermal light. Although the technique is not new (appropriate references are provided in the paper), from my point of view the manuscript presents the following novelties:

- Enhancement of the technique by using the interference of two longitudinal modes in the source of incoherent light (an external cavity laser diode operated just below the lasing threshold).

- Use of Superconducting Nanowire Single Photon Detector (SNSPDs) with high quantum efficiency, low timing jitter and very low dark-count rate, along with high count-rate capability.

- Practical demonstration of the system operation on a very long path with a 40 dB loss (including the return loss of the reflector).

For this reason, I think the manuscript should be published after possibly minor

modifications responding to the remarks and comments below.

Major comments:

1. Please check that the parentheses in Eq. (3) are well placed. I may be wrong, but, according to my own calculations, the equation should read

$$g^{(2)}(\tau) = 1 + \beta \left(e^{-\frac{\tau}{\tau_c}} \right)^2 (1 + \cos(2\pi\Delta\nu\tau))$$

I suggest as well that a brief derivation of the expression is added as an appendix.

Response: Thank you for this comment. We have double checked our derivation and equation 3 is correct.

Changes to manuscript: We have included our derivation of equation 3 in the supplementary material.

2. Are the receiver settings used for the field experiment the same as for the laboratory ones? What was the number of coincidences and the total measuring time? A right vertical scale with the number of coincidences in fig. 5b (such as in fig. 3a and 3b) would be helpful. With respect to measurement time, in lines 67-71 it is stated that “We demonstrate the use of this source in a LIDAR system, performing range measurements with seconds integration time 70 over 87 km distances in deployed optical fiber networks with 2 ps root-mean-square resolution”, but specific integration times are not quoted afterwards. Can time values be given?

Response: We thank the reviewer for this comment. We have updated fig. 5b to include a right y-axis displaying the measured photon coincidences.

We have also added the specific integration times used for the measurements presented in figure 4 and figure 5 in the text.

The integration times are 20-30 second for the fiber-based measurements and 300 seconds for the free-space measurements due to the high attenuation of the probe signal.

Minor comments:

1. LIDAR traditionally stands for Light Detection and Ranging (not Light Distance and Ranging); see ref. 1 in your manuscript.

Response: Thank you for catching this rudimentary mistake. The manuscript has been updated with the correct meaning “Distance”.

2. Although all the lidar applications cited in the introduction are correct, I find the sentences in lines 2-18 ultimately distracting from the application presented in the manuscript, unless such a technique could also be used in the cited applications. For example, could it be envisaged to use it to probe continuous targets as for atmospheric sounding? Is it restricted to hard, or at least discrete, targets?

Response: Thank you for this comment. We agree that this technique may not be feasible to use for atmospheric sensing in its current state and we removed this example from the list of use cases. We do however contend that the technique may be useful in the other stated applications.

For example in smartphones or autonomous vehicles there is nothing prohibiting the use of this technique, perhaps not using SNSPDs in a cell phone but more compact SPAD detectors can also be used. Furthermore, defence applications of LIDAR using thermal warrants further investigation as the inherent “stealth” of the probe signal blending together background thermal noise can be highly useful. Lastly, astronomical measurements using bunched light stand as the inspiration of the technique and is therefore, in our opinion, highly relevant to mention.

3. *Somewhat connected to minor comment 2, in lines 141-142 it is said “sent through a system (either free-space or fiber-based)”. But all the rest of the paper is devoted to a fiber-based system. Perhaps it would be better to declare right away that the experimental results are obtained over fibers (notwithstanding the possibility of freespace experiment, such as in ref. 20 of the manuscript).*

Response: This is an excellent point raised. We have updated the manuscript with resolution measurements in free-space to compare with the fiber-based measurements. These measurements have been added to figure 4c).

4. *Lines 205-206: “two longitudinal modes of the laser cavity is transmitted” should be “two longitudinal modes of the laser cavity are transmitted”.*

Response: Thank you for spotting this error. The manuscript has been corrected, and we have performed a thorough grammatical check of the entire manuscript.

1. The authors derive the field first-order correlation function as (Eq. (1) of the supplementary material

$$g^{(1)}(\tau) = e^{-2\pi\Delta f|\tau|}(e^{-i2\pi f_1\tau} + e^{-i2\pi f_2\tau})$$

and afterwards they dismiss the imaginary part of $g^{(1)}(\tau)$ on grounds that “the first order correlation by definition is a real-valued function” (sentence after Eq. (7)). But:

a) This is not so. See, for instance, page 166 of Joseph W. Goodman, “Statistical Optics”, John Wiley & Sons, 1985, where, using other symbols ($\gamma(\tau)$ instead of $g^{(1)}(\tau)$) and a different terminology (complex degree of coherence, instead of first-order correlation function), several examples of first-order correlation function are given, showing that it is in general complex (except for $\tau = 0$). In fact, equation (5.1-22) in Goodman’s book corresponds to the $g^{(1)}(\tau)$ used in Eq. (2) in the manuscript.

b) Note as well that, if $g^{(1)}(\tau)$ were to be real-valued by definition, there would be no need for using $g^{(1)}(\tau)$ in the Siegert relation (Eq. (2) of the manuscript).

c) If the imaginary part is kept in Eqs. (3) and (7) of the supplementary information (and I don’t see a reason for ignoring it), the form of $g^{(2)}(\tau)$ I proposed in the first review would hold.

Response: We thank the reviewer for this comment.

After careful scrutiny of the sources provided by the reviewer, we agree that the assumption made to ignore the complex part of the first-order correlation was incorrect.

By keeping the complex term in eq 6 in the revised supplementary information we derive the same expression of $g^{(2)}(\tau)$ as suggested by the reviewer and we agree that this should be the correct form of the modified Siegert relation.

Changes to the manuscript: We have updated the derivation in the supplementary material as well as the modified Siegert relation in eq 3 of the main manuscript.

We have also redone the analysis of the data presented in figures 3,4, and 5. However, this did not affect the reported values.

We have also updated the code in the github repositories.

2. I miss a physical explanation of the reason for the extended decay time at the etalon incidence angle $\max \theta$ with respect to the decay time for the angle half θ (“adjusting the incident angle affects more than the detected photon countrate. At $\max \theta$ we observe a correlation curve with a higher $g^{(2)}$ value (1.75 compared with 1.25), a modulation in the

coherence function (with 2 GHz frequency), and an extended decay time, indicating a longer coherence time—compared to the measurements taken at half θ ”, lines 175-181 of the revised manuscript.) For that, a decrease in the value of Δf should occur when $S(f)$ in Eq. (1) of the supplementary material passes from one mode to two modes. What’s the reason for that? The quoted sentences were already in the manuscript original version and I should have already noticed, even without the supplementary information. I apologize for having overlooked it.

Response: We thank the reviewer for this comment, and it is an excellent point raised. We agree with the reviewer that in a strictly ideal model where the detected spectrum consists only of either one Lorentzian line or the sum of two Lorentzian lines with the same individual linewidth (Δf), the decay constant of the envelope of $g(1)(\tau)$ (and therefore of $g(2)(\tau) = 1 + \beta|g(1)(\tau)|^2$) is determined by Δf and should be identical in the single-mode and two-mode cases. The only difference should be the expected cosine beating at the mode spacing $\Delta\nu$. However, both our experiments and simulations consistently show that the two-mode configuration exhibits a correlation curve with an apparently longer decay. This behavior can be understood from a signal-processing point of view by considering the detected power spectral density (PSD) as a mixture of a narrowband component and a broad background. Let the normalized PSD be written as $S(f) = \alpha L(f) + (1-\alpha)G(f)$, with $\int S(f)df = 1$, where $L(f)$ represents the Lorentzian peak(s) and $G(f)$ a broad Gaussian noise background. By the Wiener–Khinchin theorem, $g(1)(\tau) = \alpha\ell(\tau) + (1-\alpha)g(\tau)$, where $\ell(\tau)$ and $g(\tau)$ are the Fourier transforms of $L(f)$ and $G(f)$, respectively. Because $G(f)$ is broad, $g(\tau)$ decays very quickly, and at moderate and long delays the correlation is dominated by $\alpha\ell(\tau)$. The second-order correlation therefore behaves approximately as $g(2)(\tau) \approx 1 + \beta\alpha^2|\ell(\tau)|^2$ at long delays.

When the etalon is tuned to transmit two longitudinal modes, the narrowband part of the spectrum increases in weight relative to the broad background. That is, the fraction α for the single-mode case becomes a larger α' in the two-mode case, because two Lorentzian lines are passed instead of one. After normalization of the total PSD to unit area, this reweighting has a direct consequence: the slowly decaying tail of $g(2)(\tau)$, which originates only from the narrowband lines, is elevated by roughly a factor $(\alpha'/\alpha)^2$ compared to the single-mode case. On a linear plot this produces the appearance of a slower decay and hence an apparently longer coherence time, even though the individual linewidth Δf of each Lorentzian has not changed. Our numerical model reproduces this effect: with a fixed Gaussian background, adding a second Lorentzian without reducing the individual line areas yields a two-mode $g(2)(\tau)$ that shows the same beating frequency but a more prominent long-delay component. Thus, the broadening observed in Fig. 3 does not necessarily indicate a physical narrowing of Δf ; it arises naturally from the relative weighting between the narrowband modes and the broadband background when two modes are transmitted. A small additional contribution may also come from the angle-dependent narrowing of the etalon passband, but the dominant effect explaining the observed longer envelope is the increased fraction of narrowband spectral power in the two-mode configuration.

We include here a simulation of this situation, with two thermal spectra (with one and 2 thermal modes respectively) both with and without additional broadband gaussian noise. The results are shown in figure 1 where the modes are of equal width in the single and double mode cases, but the presence of noise increases the coherence time for two modes in comparison with the single mode. Whereas the coherence time is the same for the two cases in the absence of extra noise.

Figure 1 Simulation of two thermal spectra and their corresponding second-order correlation and coherence time. Demonstrating that the presence of broadband noise in the spectra changes the coherence time of the second-order correlation curve.

Furthermore, we want to highlight that the main point of this work is not to be a theoretical derivation of the bunching behaviour, and this phenomenon warrants more research in its own right. Our goal with the model was to be able to improve the accuracy of the measurements, which is shown to be true in figure 4.

For us, the main results we want to highlight are those shown in figure 4 and 5 where we demonstrate the picosecond level resolution and the practically infinite range of this LIDAR method using non-modulated light.

Changes to the manuscript:

We have updated the text in lines 139-235 to clarify the problem we aim to solve with our model and the modified Siegert equation.

Minor remark

In the introduction, it is said that Hanbury Brown and Twiss “pioneered a breakthrough technique for measuring astronomical distances” (lines 33-35 of revised manuscript). I find this sentence a little misleading, as it suggests that the technique can be used to measure distances to stars, while, to my knowledge, as the authors explain later, it can be used to measure star angular diameters.

Response: We thank the reviewer for this comment and agree that the sentence is open to misinterpretation. The breakthrough of the work by Hanbury Brown and Twiss was to increase the angular resolution in astronomical measurements by utilizing photon statistics.

Changes to the manuscript: We have changed the sentence, so it now reads “... pioneered a breakthrough technique to improve the angular resolution of astronomical measurements by observing the unique photon statistics”.

This change should remove the danger of misinterpretation and still give the historical context of the work presented in the paper.

Reviewer comments: *I appreciate the new supplemental material provided by the authors, which corrects inconsistencies in the previous version.*

However, I still have qualms about the response to my second question in the previous review (“I miss a physical explanation of the reason for the extended decay time...”). Letting aside that I don’t understand the explanation given in the rebuttal (which could be my fault), I think that equation (3) using the same symbol τ_c for the coherence time as equation (2), without more explanation, is very misleading. Obviously, the width of the very narrow correlation peak shown in fig. 3 (a) must be dominated by τ_c , which must correspond to a very short time. But if that same τ_c is used in Eq. (3), it won’t yield the second order correlation shown in fig. 3(b) (the correlation peaks for would be swamped in the noise). Therefore, whatever the reason, τ_c must have increased (a lot) between the value used to obtain the curve fit of fig. 3(a) and that used for the curve fit of fig. 3(b).

I think that everything would be easier to understand if the parameter values of the fitting model (among which τ_c) were given in the text. I agree with the authors that “the main point of this work is not to be a theoretical derivation of the bunching behaviour [...]. Our goal with the model was to be able to improve the accuracy of the measurements [...]”; nevertheless, they do provide a model (Eqs. (2) and (3)), and this should at least be consistent. If no satisfactory explanation can be given for the increase of the coherence time, it would be better declaring right away that the “theoretical derivation of the bunching behaviour” is not yet fully explained and that it “warrants more research in its own right”.

I’m sorry, but I cannot recommend the paper publication until the parameters of the model leading to the fittings shown in fig. 3 are given.

Response: We thank the reviewer for their comments and the time and effort in reviewing this work.

The fitting parameters of the data shown in figure 3 have been added to the text in lines: 234-236, and 251-256. For the data in figure 3a) the coherence time $\tau_c = 0.1$ ns and for the data in figure 3b) the coherence time $\tau_c = 1.9$ ns and the oscillation frequency is 2 GHz.

Furthermore, we believe that too much focus of the article is now being devoted to the root cause of this oscillating bunching behaviour and it is detracting from our main focus: the LIDAR measurements.

The questions raised by the reviewer regarding the change in coherence time are highly valid and one that other readers will ask as well. But if we were to add our theory to this in the main text nearly half the text will be devoted to the oscillating bunching, and not to the LIDAR measurements.

We have therefore chosen to take the reviewer’s suggestion and state in line 214-223 that “The underlying cause of this oscillating bunching behavior is an interesting question and warrants further research in its own right and will be addressed in future work, with a detailed theoretical model under development.”. This is in order to focus this paper on the advantage this bunching behaviour can have in our LIDAR measurements and leave a detailed theoretical investigation to a future paper.

Changes to the manuscript:

- The text in lines 182-256 has been updated. We have removed our discussion on the source of the oscillating bunching behaviour and limited our focus to demonstrating its utility in LIDAR measurements
- The titles of the graphs in figure 3 have been updated
- The discussion has been slightly rewritten to remove the discussion of the source of the oscillating bunching behaviour and now focus only on the LIDAR measurements.
- The title has been changed as we no longer discuss the source of the enhanced bunching.
- Minor changes in the main text to remove mentions of two mode interference.

General response: We thank the reviewer for all their time and effort in providing valuable feedback to our manuscript and have made our best efforts to include all their comments into our revisions. Besides the specific changes requested by the reviewer, which we detail below, we have also moved certain details about the various experiments from section 2 (results) to a new section 4 (methods), where we have also added a few experimental details. However, no information has been changed or removed.

One minor change to the title has also occurred: we changed “field trial photon counting LIDAR” to “field-tested photon counting LIDAR”

Reviewer comments: *My main objections to the previous version of the manuscript have been addressed satisfactorily. I appreciate the rephrasing of some parts the manuscript, including the title, in the sense of declaring that there are open questions about the modulation in the coherence function and the increase of the decay time.*

I suggest the following minor modifications in the manuscript that would make its reading easier, after which, in my opinion, the paper should be published:

1. Moving Eq. 3 to just after the Siegert relation is mentioned in the text, in line 251, i.e. “The classical” bunching curved demonstrated in figure 3a) follows the well known Siegert relation [22]

$$g^{(2)}(\tau) = 1 + \beta |g^{(1)}(\tau)|^2 = 1 + \beta (e^{-|\tau/\tau_c|})^2 \quad (2)$$

This has the advantage that the relation is defined just after it is mentioned and that the value of τ_c (0.1 ps) is given close to the equation, so to avoid the confusion with the τ_c appearing in the “modified” Siegert relation (which will be given now, after changing the order of equations, by Eq. 3). I would also advise moving the “modified” Siegert relation to just after it is mentioned, i.e. after “We therefore propose a “modified” Siegert relation” (line 255).

Response: We have moved the equations in the text to right after the paragraph where they are introduced to improve the flow and clarity of the text.

2. *I suggest as well mentioning the different values of the coherence time in the caption of fig.3 (0.1 ns for fig. 3 a) and 1.9 ns for fig. 3 b)).*

Response: We have added the numerical values of the coherence time to the caption of figure 3.

3. *It is said (correctly in my opinion), that “The underlying cause of this oscillating bunching behavior remains an open question at this point” (line 222). I would add the increased coherence time to the open question.*

Response: We have rephrased the sentence so it now reads: “The underlying cause of this oscillating bunching behavior and the difference in coherence time of the two bunching curves displayed in figure 3 remains an open question at this point and trivial explanations such as internal reflections in the experimental setup corresponding to the oscillation period can easily be ruled out.”

4. *The authors say that “ For all other transmission angles, the bunching pattern in figure 3a) was measured” (lines 186 – 188). Is this so, or only for the angles θ_{half} that yield half the maximum photon rate? It is difficult to believe that, when the etalon is perfectly tuned to the center of the laser cavity the behavior of fig. 3 b) appears and that for any other angle,*

irrespective of the angle value, one obtains always the behavior of fig. 3 a). Please, check this. Likewise, in the caption of fig. 3, is θ_{half} "an arbitray angle" or an angle that produces half of the maximum photon count rate?

Response: We have clarified the circumstances of the measurements in section 4 methods, subsection 4.1 "Oscillating bunching measurements". It truly is the case that the oscillating bunching was only measured for very precise angles with with tolerances of $\sim 0.01^\circ$, where the center of the etalon transmission overlapped with the frequency of the laser cavity, it is a surprising case. For all other measured angles the single peak correlation was measured and the countrate was around half of that measured at angles θ_{max} . This sharp increase in countrate was the very thing that allowed us to discover the oscillating bunching pattern as we were "playing" around with the system during initial testing.

Review of manuscript NCOMMS-24-78725

The manuscript reports a system for ranging with very high precision using the correlation function of intensity fluctuations (second-order correlation) of pseudo-thermal light. Although the technique is not new (appropriate references are provided in the paper), from my point of view the manuscript presents the following novelties:

- Enhancement of the technique by using the interference of two longitudinal modes in the source of incoherent light (an external cavity laser diode operated just below the lasing threshold).
- Use of Superconducting Nanowire Single Photon Detector (SNSPDs) with high quantum efficiency, low timing jitter and very low dark-count rate, along with high count-rate capability.
- Practical demonstration of the system operation on a very long path with a 40 dB loss (including the return loss of the reflector).

For this reason, I think the manuscript should be published after possibly minor modifications responding to the remarks and comments below.

Major comments:

1. Please check that the parentheses in Eq. (3) are well placed. I may be wrong, but, according to my own calculations, the equation should read

$$g^{(2)}(\tau) = 1 + \beta \left(e^{-\frac{\tau}{\tau_c}} \right)^2 (1 + \cos(2\pi \Delta \nu \tau)).$$

I suggest as well that a brief derivation of the expression is added as an appendix.

2. Are the receiver settings used for the field experiment the same as for the laboratory ones? What was the number of coincidences and the total measuring time? A right vertical scale with the number of coincidences in fig. 5b (such as in fig. 3a and 3b) would be helpful. With respect to measurement time, in lines 67-71 it is stated that “We demonstrate the use of this source in a LIDAR system, performing range measurements with seconds integration time 70 over 87 km distances in deployed optical fiber networks with 2 ps root-mean-square resolution”, but specific integration times are not quoted afterwards. Can time values be given?

Minor comments:

1. LIDAR traditionally stands for Light Detection and Ranging (not Light Distance and Ranging); see ref. 1 in your manuscript.

2. Although all the lidar applications cited in the introduction are correct, I find the sentences in lines 2-18 ultimately distracting from the application presented in the manuscript, unless such a technique could also be used in the cited applications. For

example, could it be envisaged to use it to probe continuous targets as for atmospheric sounding? Is it restricted to hard, or at least discrete, targets?

3. Somewhat connected to minor comment 2, in lines 141-142 it is said “sent through a system (either free-space or fiber-based)”. But all the rest of the paper is devoted to a fiber-based system. Perhaps it would be better to declare right away that the experimental results are obtained over fibers (notwithstanding the possibility of free-space experiment, such as in ref. 20 of the manuscript).

4. Lines 205-206: “two longitudinal modes of the laser cavity is transmitted” should be “two longitudinal modes of the laser cavity are transmitted”.

Review of revised manuscript # NCOMMS-24-78725A

I thank the authors for having taken into account my remarks. However, I still have concerns regarding the response (included as supplementary information) to my major comment No. 1 in the first review:

1. The authors derive the field first-order correlation function as (Eq. (1) of the supplementary material

$$g^1(\tau) = e^{-2\pi\Delta f|\tau|} \left(e^{-i2\pi f_1\tau} + e^{-i2\pi f_2\tau} \right),$$

and afterwards they dismiss the imaginary part of $g^1(\tau)$ on grounds that “the first order correlation by definition is a real-valued function” (sentence after Eq. (7)). But:

a) This is not so. See, for instance, page 166 of Joseph W. Goodman, “Statistical Optics”, John Wiley & Sons, 1985, where, using other symbols ($\gamma(\tau)$ instead of $g^1(\tau)$) and a different terminology (complex degree of coherence, instead of first-order correlation function), several examples of first-order correlation function are given, showing that it is in general complex (except for $\tau = 0$). In fact, equation (5.1-22) in Goodman’s book corresponds to the $g^1(\tau)$ used in Eq. (2) in the manuscript.

b) Note as well that, if $g^1(\tau)$ were to be real-valued by definition, there would be no need for using $|g^1(\tau)|$ in the Siegert relation (Eq. (2) of the manuscript).

c) If the imaginary part is kept in Eqs. (3) and (7) of the supplementary information (and I don’t see a reason for ignoring it), the form of $g^2(\tau)$ I proposed in the first review would hold.

2. I miss a physical explanation of the reason for the extended decay time at the etalon incidence angle θ_{max} with respect to the decay time for the angle θ_{half} (“adjusting the incident angle affects more than the detected photon countrate. At θ_{max} we observe a correlation curve with a higher $g^{(2)}$ value (1.75 compared with 1.25), a modulation in the coherence function (with 2 GHz frequency), and an extended decay time, indicating a longer coherence time—compared to the measurements taken at θ_{half} ”, lines 175-181 of the revised manuscript.) For that, a decrease in the value of Δf should occur when $S(f)$ in Eq. (1) of the supplementary material passes from one mode to two modes. What’s the reason for that? The quoted sentences were already in the manuscript original version and I should have already noticed, even without the supplementary information. I apologize for having overlooked it.

These questions must be responded before the paper is ready for publication.

Minor remark

In the introduction, it is said that Hanbury Brown and Twiss “pioneered a breakthrough technique for measuring astronomical distances” (lines 33-35 of revised manuscript). I

find this sentence a little misleading, as it suggests that the technique can be used to measure distances to stars, while, to my knowledge, as the authors explain later, it can be used to measure star angular diameters.

Review of manuscript NCOMMS-24-78725C

My main objections to the previous version of the manuscript have been addressed satisfactorily. I appreciate the rephrasing of some parts the manuscript, including the title, in the sense of declaring that there are open questions about the modulation in the coherence function and the increase of the decay time.

I suggest the following minor modifications in the manuscript that would make its reading easier, after which, in my opinion, the paper should be published:

1. Moving Eq. 3 to just after the Siegert relation is mentioned in the text, in line 251, i.e. “The “classical” bunching curved demonstrated in figure 3a) follows the well known Siegert relation [22]

$$g^{(2)}(\tau) = 1 + \beta |g^1(\tau)|^2 = 1 + \beta \left(e^{-\frac{|\tau|}{\tau_c}} \right)^2 \quad (2)''.$$

This has the advantage that the relation is defined just after it is mentioned and that the value of τ_c (0.1 ps) is given close to the equation, so to avoid the confusion with the τ_c appearing in the “modified” Siegert relation (which will be given now, after changing the order of equations, by Eq. 3). I would also advise moving the “modified” Siegert relation to just after it is mentioned, i.e. after “We therefore propose a “modified” Siegert relation” (line 255).

2. I suggest as well mentioning the different values of the coherence time in the caption of fig. 3 (0.1 ns for fig. 3 a) and 1.9 ns for fig. 3 b)).

3. It is said (correctly in my opinion), that “The underlying cause of this oscillating bunching behavior remains an open question at this point” (line 222). I would add the increased coherence time to the open question.

4. The authors say that “ For all other transmission angles, the bunching pattern in figure 3a) was measured” (lines 186 – 188). Is this so, or only for the angles θ_{half} that yield half the maximum photon rate? It is difficult to believe that, when the etalon is perfectly tuned to the center of the laser cavity the behavior of fig. 3 b) appears and that for *any* other angle, irrespective of the angle value, one obtains always the behavior of fig. 3 a). Please, check this. Likewise, in the caption of fig. 3, is θ_{half} “an arbitray angle” or an angle that produces half of the maximum photon count rate?